# Intracellular Toxic Advanced Glycation End-Products in 1.4E7 Cell Line Induce Death with Reduction of Microtubule-Associated Protein 1 Light Chain 3 and p62

**DOI:** 10.3390/nu14020332

**Published:** 2022-01-13

**Authors:** Takanobu Takata, Akiko Sakasai-Sakai, Masayoshi Takeuchi

**Affiliations:** Department of Advanced Medicine, Medical Research Institute, Kanazawa Medical University, Kahoku 920-0293, Ishikawa, Japan; asakasai@kanazawa-med.ac.jp (A.S.-S.); takeuchi@kanazawa-med.ac.jp (M.T.)

**Keywords:** glyceraldehyde (GA), advanced glycation end-products (AGEs), toxic AGEs (TAGE), type 1 diabetes mellitus (T1DM), type 2 diabetes mellitus (T2DM), pancreatic islet β-cells (β-cells), microtubule-associated protein 1 light chain 3 (LC3), p62

## Abstract

Background: The death of pancreatic islet β-cells (β-cells), which are the insulin-producing cells, promote the pathology in both Type 1 and Type 2 diabetes mellitus (DM) (T1DM and T2DM), and they are protected by autophagy which is one of the mechanisms of cell survival. Recently, that some advanced glycation end-products (AGEs), such as methylglyoxial-derived AGEs and *N*^ε^-carboxymethyllysine, induced the death of β-cells were revealed. In contrast, we had reported AGEs derived from glyceraldehyde (GA, the metabolism intermediate of glucose and fructose) are considered to be toxic AGEs (TAGE) due to their cytotoxicity and role in the pathogenesis of T2DM. More, serum levels of TAGE are elevated in patients with T1 and T2DM, where they exert cytotoxicity. Aim: We researched the cytotoxicity of intracellular and extracellular TAGE in β-cells and the possibility that intracellular TAGE were associated with autophagy. Methods: 1.4E7 cells (a human β-cell line) were treated with GA, and analyzed viability, quantity of TAGE, microtubule-associated protein 1 light chain 3 (LC3)-I, LC3-II, and p62. We also examined the viability of 1.4E7 cells treated with TAGE-modified bovine serum albumin, a model of TAGE in the blood. Results: Intracellular TAGE induced death of 1.4E7 cells, decrease of LC3-I, LC3-II, and p62. Extracellular TAGE didn’t show cytotoxicity in the physiological concentration. Conclusion: Intracellular TAGE induced death of β-cells more strongly than extracellular TAGE, and may suppress autophagy via reduction of LC3-I, LC3-II, and p62 to inhibit the degradation of them.

## 1. Introduction

Diabetes mellitus (DM) exists as type 1 (T1DM) and type 2 (T2DM). The latter is a lifestyle-related disease (LSRD) and one of the greatest public health challenges in the world [1]. The death of pancreatic islet β-cells (β-cells), which are the insulin-producing cells, promote the pathology in both T1 and T2DM because the lack of production and secretion of insulin were induced [2,3]. The death and dysfunction of β-cells result in glucotoxicity, lipotoxicity, and oxidative stress [4,5]. In contrast, there is a system that protects β-cells against dysfunction and death such as autophagy in T2DM [4]. Recently, advanced glycation end-products (AGEs), such as metylglyoxial-derived AGEs (MGO-AGEs) and *N*^ε^-carboxymethyllysine (CML), have been reported to be generated/exist in β-cells, induce death and dysfunction of them, and they may be associated with the promotion of DM [6,7]. On the other hand, AGEs derived from glyceraldehyde (GA), an intermediate of glucose and fructose metabolism, have been named toxic AGEs (TAGE) because of their cytotoxicity and role in the pathogenesis of LSRD, including T2DM [1,2,8]. Previously, we showed that serum TAGE levels were elevated in patients with T1 and T2DM, which correlated with inflammatory biomarkers and the soluble form of the receptor for AGEs (RAGE) [2]. We also demonstrated the leakage of intracellular organ-generated TAGE into the blood (extracellular TAGE), where it exerted cytotoxicity [1,2,8]. It remains unclear whether TAGE induces β-cell death or dysfunction; however, intracellular TAGE may be generated in β-cells, because the pathways that produce GA from glucose and fructose are functional in these cells [1,8,9,10,11], and induce cell death, similar to observations in other cell types such as hepatocyte [12,13], cardiomyocyte [14], pancreatic ductal cell [15], skeletal muscle myoblast cell [16], and neuroblastoma cell [17].

In the present study, we investigated whether intracellular TAGE could induce cell death in 1.4E7 cells (a cell line of human β-cells [18,19]). We hypothesized that reduction of microtubule-associated protein 1 light chain 3 (LC3)-II which is produced from LC3-I, but not p62 is associated with the death of 1.4E7 cells, in a manner similar to cardiomyocytes that generate intracellular TAGE [14]. Since LC3-II and p62 are needed to generate autophagosomes to collect the unnecessary and toxic proteins, autophagy flux was regulated by them, and the increase/decrease of protein levels of them was an appropriate index of the induction/suppression of autophagy [14]. We examined the viability, generation of intracellular TAGE, and LC3-I, LC3-II, and p62 protein levels in 1.4E7 cells treated with GA. Since we hypothesized that circulating TAGE induced cytotoxicity in β-cells via RAGE [1,7,20,21], we assessed the viability of 1.4E7 cells treated with TAGE-modified bovine serum albumin (BSA) (TAGE-BSA), a model of TAGE in the blood.

## 2. Materials and Methods

### 2.1. Materials

Roswell Park Memorial Institute (RPMI)-1640 culture medium and penicillin-streptomycin solution were purchased from Sigma-Aldrich (Saint Louis, MO, USA). GA was purchased from Nacalai Tesque Inc. (Kyoto, Japan). The Cell Titer Glo assay kit was from Promega (Madison, WI, USA). KAC Co., Ltd. (Kyoto, Japan) provided the 1.4E7 cells. A horseradish peroxidase (HRP)-linked molecular weight size marker was purchased from Bionexus (Oakland, CA, USA; #BNPM41). All other reagents and kits were obtained from Fujifilm Wako Pure Chemical Co. (Osaka, Japan). TAGE-BSA, non-glycated BSA (NG-BSA), and the anti-TAGE antibody were prepared as previously reported [22].

### 2.2. Cell Culture

The condition of the culture of cells was decided as previously described with some modification [15,16]. Cells (1.4E7) were incubated in RPMI-1640 culture medium containing 5.6 mM glucose, 10% fetal bovine serum, 100 U/mL penicillin, and 100 μg/mL streptomycin, under standard cell culture.

### 2.3. GA and Aminoguanidine (AG) Treatments

Cells were seeded in every 5 wells. The conditions of GA and AG treatment were as described with some modifications [14,15,16]. Experiments were conducted 24 h after the cells were treated with 0, 1.5, 2, 2.5, and 3 mM GA. Cells were cultured in the same way before pretreating with AG, an inhibitor of AGE production. Cells were pretreated with 0 or 16 mM AG for 2 h and then exposed to 0, 2, or 3 mM GA for 24 h.

### 2.4. Viability of 1.4E7 Cells Treated with GA and AG

To cell assess the viability, adenosine triphosphate (ATP) levels were measured with CellTiter-Glo assay. The CellTiter-Glo luminescent assay kit was used, according to the manufacturer’s instructions (Promega).

### 2.5. Measurement of Intracellular TAGE Levels in 1.4E7 Cells Treated with GA and AG Using a Slot Blot (SB) Analysis

Cells were lysed in buffer [a solution of 1.8 M thiourea, 6.3 M urea, 3.6% 3-[(3-cholamido-propyl)-dimethyl-ammonio]-1-propane sulfonate), and 27 mM Tris] containing protease inhibitor [14,15]. SB analysis was performed as previously described [14].

### 2.6. Western Blot (WB) Analysis of LC3, p62, and GAPDH

WB analysis was performed as previously described, with some modifications [14,15]. Cell lysates (protein mass was 10 μg) were loaded onto a 4–15% polyacrylamide gradient gel. To analyze LC3 levels, proteins were incubated on PVDF membranes with rabbit polyclonal anti-LC3 (Cell Signaling Technology, Danvers, MA, USA; #12741, 1:1000) as the primary antibody. The PVDF membranes were incubated overnight at 4°C, washed four times, and incubated with a secondary HRP-linked anti-rabbit IgG antibody (Thermo Fisher Scientific Inc., Waltham, MA, USA; #31458, 1:3000) at room temperature for 1 h. Membranes were washed, immunoreactive proteins were detected, followed by SB analysis. The primary and secondary antibodies were removed from the PVDF membranes using WB stripping solution. Membranes were then washed and blotted with rabbit monoclonal [EPR18351] anti-p62 antibody (Abcam, Cambridge, UK; ab207305, 1:80,000), anti-GA-3 phosphate dehydrogenase (GAPDH) antibody (Abcam; ab8243, 1:10,000), HRP-linked anti-rabbit IgG antibody (1:3000), and HRP-linked anti-mouse IgG antibody (#31432,1:5000). To analyze p62 and GAPDH proteins. PVDF membranes were incubated with anti-p62 and anti-GAPDH antibodies at room temperature for 2 h, followed by HRP-linked anti-rabbit IgG antibody and HRP-linked anti-mouse IgG antibody at room temperature for 1 h.

### 2.7. Treatment of 1.4E7 Cells with NG-BSA and TAGE-BSA and Evaluation of Cell Viability

Cells were seeded in every 5 wells. The conditions of TAGE-BSA and NG-BSA treatment were described with some modifications [15,16]. 1.4E7 cells were treated with 0, 10, 20, 50, 100, and 200 μg/mL of NG-BSA and TAGE-BSA, and maintained for 24 h. To cell assess the viability, ATP levels were measured with CellTiter-Glo assay.

### 2.8. Statistical Analysis

The cell viability assay, SB analysis, and WB analysis were performed in three independent experiments. Stat Flex software (which version is 6) was used to perfume multiple comparisons (Artech Co., Ltd., Osaka, Japan). Data are shown as the mean ± standard deviation (S.D.). Significant differences in the means of each group were examined via one-way analysis of variance and Tukey’s test. Statistical significance was set at *p* < 0.05.

## 3. Results

### 3.1. Cell Viability and Intracellular TAGE Levels in GA- and AG-Treated 1.4E7 Cells

An approximately dose-dependent decrease in the viability of 1.4E7 cells was observed following the addition of GA (Figure 1a). Similarly, intracellular TAGE levels in 1.4E7 cells increased with GA in an approximately dose-dependent manner (Figure 1b). A linear GA dose-dependent response was not observed. The viability of 1.4E7 cells decreased dose-dependently to 52 and 9% following the addition of 2 and 3 mM GA, respectively, without AG, an inhibitor of AGE production (Figure 1c). AG without GA induced cell death. Therefore, the viability of 1.4E7 cells pretreated with 16 mM AG was 65, 65, and 60% following the addition of 0, 2, and 3 mM GA, respectively, and no significant difference was observed between groups, AG showed the complete inhibition of decrease of cell viability. In the absence of AG pretreatment, intracellular TAGE levels dose-dependently increased following GA treatment; however, these increases were completely inhibited in 1.4E7 cells pretreated with AG (Figure 1d).

### 3.2. Protein Levels of LC3-I, LC3-II, and p62 in 1.4E7 Cells Treated with GA and AG

LC3-I, LC3-II, and p62 protein levels in 1.4E7 cells that generated intracellular TAGE revealed three patterns (Figure 2 and Appendix A Appendix A). In the first pattern, protein levels of LC3-I, LC3-II, and p62 were maintained at normal levels. In the second pattern, protein levels of p62 were reduced. In the third pattern, the levels of all proteins were reduced.

### 3.3. Effects of AG Pretreatment on LC3-I, LC3-II, and p62 Protein Levels in GA-Treated 1.4E7 Cells

Treatment with 3 mM GA without AG decreased LC3-I and LC3-II protein levels to 34 and 25%, respectively (Figure 3a–c, Appendix A Appendix A). These effects were inhibited following pretreatment with AG. Treatment with 2 or 3 mM GA, without AG, resulted in a dose-dependent decrease in the protein levels of p62 in 1.4E7 cells (Figure 3a,d, Appendix A Appendix A), while AG pretreatment completely inhibited this decrease. In contrast, AG treatment without GA did not affect the protein levels of them (Figure 3 and Supplementary Material Appendix A).

### 3.4. Viability of 1.4E7 Cells Treated with NG-BSA and TAGE-BSA

There were no significant differences in the viability of 1.4E7 cells following treatment with 10, 20, or 50 μg/mL of NG-BSA and TAGE-BSA (Figure 4a–c). The viability of 1.4E7 cells following treatment with 100 μg/mL NG-BSA and TAGE-BSA was 102 and 94%, respectively, with a cell viability ratio (TAGE-BSA/NG-BSA) of 92% (Figure 4d). The viability of cells treated with 200 μg/mL NG-BSA and TAGE-BSA was 102 and 92%, respectively, with a cell viability ratio of 90% (Figure 4e).

## 4. Discussion

In the present study, 1.4E7 cells were treated with GA to induce the rapid production of TAGE over 24 h. In a previous study, the treatment of pancreatic islets with 20 mM glucose resulted in the generation of 0.025 pmol GA per islet, while treatment with 10 mM GA resulted in the generation of 0.12 pmol GA per islet [23]. Takahashi et al. considered that 2 mM GA corresponded to 20 mM glucose against pancreatic islet [24]. In contrast, plasma glucose levels in patients with diabetic ketoacidosis are 89.7 ± 40.1 mM [14]. Since the addition of 2 mM GA corresponded to one of 20 mM glucose, 1.4E7 cells (which were incubated in the medium containing 5.6 mM glucose) treated with 2 mM GA might be similar to exposure 25.6 mM glucose. This concentration of glucose is within the range of physiological conditions [14]. Though we have no data of the value of glucose, which is corresponded to 2.5 and 3 mM GA, 1.4E7 cells were treated with them because treatment of 4 mM GA for PANC-1cells which is one of the pancreatic ductal cell lines were examined in our previous investigation [15].

We first proved that intracellular TAGE was generated in the 1.4E7 cells treated with GA and induced death of them based on the experiment which was the treatment of GA and AG (Figure 1). There is a report that AG is capable of inhibiting enzymes that possess carbonyl groups as cofactors, such as nitric oxide synthase [25]. Since AG is the compound that induced cytotoxicity, cell viability decreased in the condition of AG without GA [16]. However, the cell viability of 1.4E7 cells which were treated with GA and AG was recovered until the same level, which was treated only AG, we can recognize that AG inhibited the generation of intracellular TAGE and death of 1.4E7 cells. Though autoimmune systems induce insulitis and the death of β-cells in T1DM, the mechanisms of death of β-cells in T2DM remain unclear [4,7,26,27]. Nevertheless, we recognize that cell death in β-cells dramatically promotes the pathology in both T1 and T2DM because the production and secretion of insulin is impossible, and the number of β-cells hardly proliferates when they are decreased by autoimmune systems or other factors [5]. Though intracellular TAGE in β-cells may be able to induce autoimmune system via other cells such as T1DM, it remains unclear in this study. Although these findings are unable to be directly transported in vivo because intracellular TAGE has not been observed in β-cells in DM patients or model animals, our results suggested the possibility that the induction of β-cell death through the generation of intracellular TAGE may more strongly promote the pathology of DM than insulin resistance and a reduction of insulin secretion [4,7,26,27]. To protect the β-cells, the lifestyle which intracellular TAGE generate and accumulate in β-cells may be avoided.

On the other hand, we focused the reason which cell viability and the generation of intracellular TAGE didn’t show decreased and increased with the completely linear GA-dose dependence though TAGE were generated via non-enzymic reaction between GA and proteins. It is possible that mechanisms of degradation of TAGE (e.g., autophagy) might exist in 1.4E7 cells, and contribute to inhibition of cell death.

Autophagy is a process of intracellular degradation, which contributes to removing unnecessary and toxic proteins [28,29]. The induction of autophagy is characterized by increased LC3-II and decreased p62 protein levels, whereas the suppression of autophagy is characterized by the opposite expression patterns [28,29]. We previously reported the possibility that intracellular TAGE suppressed autophagy in cardiomyocytes based on that the ratio of LC3-II/LC3-I decreased without change of p62 protein level [14]. In contrast, we first reported that the reduction of the protein levels of LC3-I, LC3-II, and p62 via two steps was observed in β-cells in vitro (Figure 2 and Figure 3, and Appendix A Appendix A). Recently, Drawish et al. reported a similar phenomenon in β-cells in streptozotocin-induced T1DM (STZ-T1DM) BALB/c albino mice with WB analysis, but the detection of LC3-I and LC3-II in their data are unable to be distinguished [30]. Therefore, the reduction of each protein levels of LC3-I and LC3-II remain unclear in their reports. On the other hand, Zhao et al. provided each image of LC3-I, LC3-II, p62, and kelch-like ECH-associated protein 1 (keap1) which contribute to collecting unnecessary and toxic proteins to perform autophagy in testicular cells in STZ-T1DM FVB mice with WB analysis [31]. Each image of LC3-I, LC3-II, and p62 was similar to one of our results. However, Zhao et al. quantified the protein levels of LC3-II, p62, and keap1 didn’t quantify one of LC3-I. Based on their finding, that LC3-II and p62 decreased, and keap1 increased were reported. Though they suggested that the reduction of p62-dependent autophagic degradation of keap1 was observed, the mechanisms of reduction of LC3-II and p62 didn’t be described [31]. Therefore, we considered the phenomena in our experiments are rare and interesting, and the possibility that the production of LC3-I and p62 was suppressed and/or the degradation of them was induced by intracellular TAGE. In this case, LC3-II might decrease as well as LC3-I because it is produced from LC3-I [14], and also, intracellular TAGE might inhibit even this step. Then, autophagy is suppressed via inhibition of the generation of autophagosome. This is an interesting possibility because intracellular TAGE may suppress autophagy though they may be degraded by autophagy as the toxic proteins.

On the other hand, we previously detected serum TAGE levels of approximately 11–16 U/mL in the T1 and T2DM patients, which are equivalent to 11–16 μg/mL TAGE-BSA [32,33,34,35]. TAGE-BSA, which is the physiological level, could not induce cell death. Since only more than five-fold concentration against physiological levels of TAGE-BSA induced cell death, these results suggested that circulating TAGE at physiological concentrations may not be able to induce β-cell death in patients with DM. However, studies have reported that extracellular AGEs induce cytotoxicity against β-cells when combined with environmental stress [21,36].

Our study had several limitations. First, we were unable to confirm that the phenotype of the 1.4E7 cells reflected that of primary β-cells. Although primary human or animal β-cells may be suitable for such studies, they are not commercially available and are obtained by harvesting from human donors or sacrificed animals. Though the rat primary pancreatic islets which are maintained in the flask/tube kit are commercially available, their components are not only β-cells. Therefore, we believed that the cell line used was sufficient for the analyses performed in this study. Second, although the mechanisms of cell death (e.g., necrosis and apoptosis) may be important, we analyzed cell viability by determining cellular ATP levels. Proteins associated with cell death may be TAGE-modified proteins [15,17]; TAGE-modified proteins and their effects on cell death processes cannot be studied in the short term though they are important [37]. This theme may be suitable for an independent investigation. Finally, data were only obtained for LC3-I, LC3-II, and p62 as autophagy-associated proteins. Although other proteins may be analyzed to investigate autophagy flux and signaling pathways, these proteins may be TAGE-modified and cannot be analyzed over the short term. This theme may also be suitable as an independent investigation containing the mechanisms that reduction of LC3-I, LC3-II, and p62 didn’t show a linear GA-dose dependence.

Collectively, the present results showed that 1.4E7 cells, which are the cell lines of insulin-producing cells generated intracellular TAGE, which induced cell death more strongly than extracellular TAGE. Intracellular TAGE may suppress autophagy via reduction of LC3-I, LC3-II, and p62 to inhibit the degradation of them.

## Figures and Tables

**Figure 1 nutrients-14-00332-f001:**
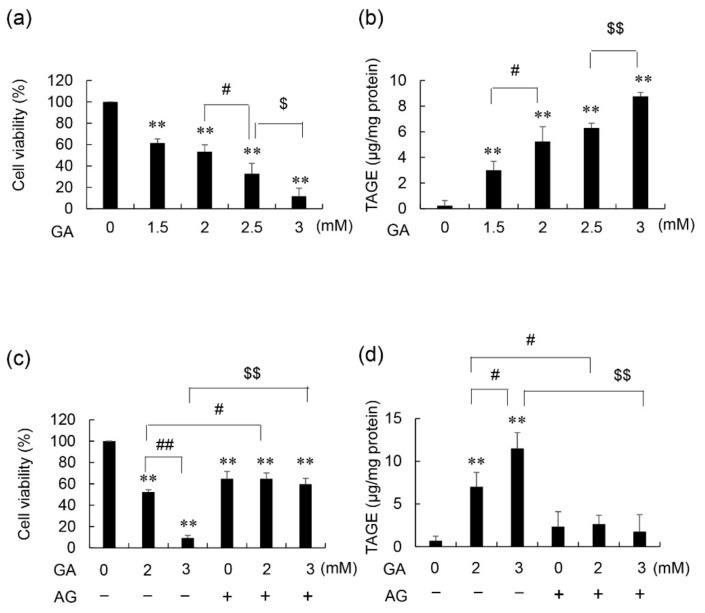
Viability and intracellular TAGE levels in glyceraldehyde (GA)- and aminoguanidine (AG)-treated 1.4E7 cells. (**a**,**b**) Effect of treatments with 0–3 mM GA for 24 h. (**c**,**d**) Effects of pretreatments with 0 or 16 mM AG for 2 h, followed by 0–3 mM GA for 24 h. (**a**,**c**) Cells were seeded in five wells and the average was calculated. The CellTiter-Glo luminescent assay was performed to evaluate cell viability in three independent experiments. Data are shown as the mean ± S.D. (*n* = 3). (**b**,**d**) A slot blot (SB) analysis was performed to assess intracellular TAGE. Cell lysates (Two μg of proteins per lane) were blotted onto polyvinylidene difluoride (PVDF) membranes. TAGE levels were calculated using a calibration curve of TAGE-modified bovine serum albumin (TAGE-BSA). The SB analysis was conducted in three independent experiments. One experiment was performed using two lanes and the average was calculated. Data are shown as the mean ± S.D. (*n* = 3). *p*-values were based on Tukey’s test. (**a**) ** *p* < 0.01 vs. 0 mM GA. ^#^
*p* < 0.05 vs. 2 mM GA. ^$^
*p* < 0.05 vs. 2.5 mM GA. (**b**) ** *p* < 0.01 vs. 0 mM GA. ^#^
*p* < 0.05 vs. 1.5 mM GA. ^$$^
*p* < 0.01 vs. 2.5 mM GA. (**c**,**d**) ** *p* < 0.01 vs. 0 mM GA without AG. ^#^
*p* < 0.05 vs. 2 mM GA without AG. ^##^
*p* < 0.01 vs. 2 mM GA without AG. ^$$^
*p* < 0.01 vs. 3 mM GA without AG.

**Figure 2 nutrients-14-00332-f002:**
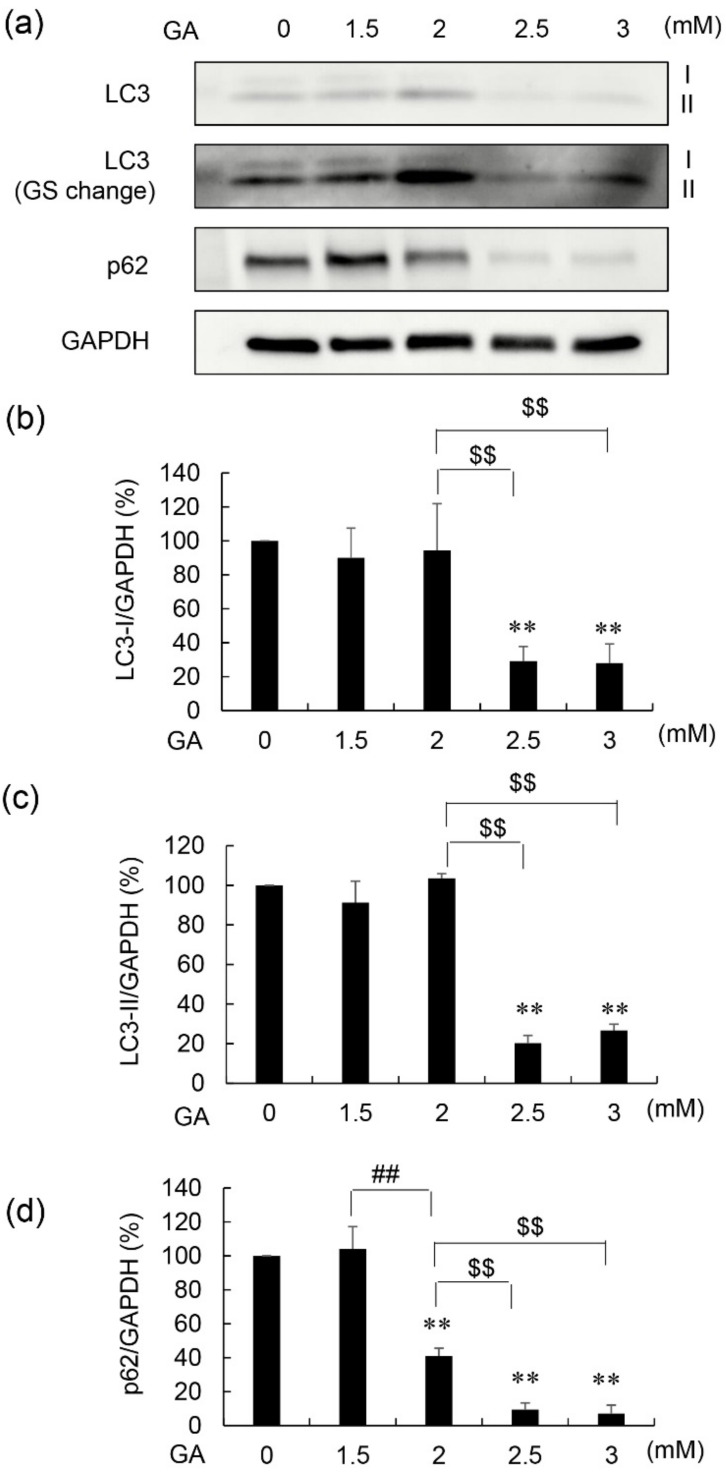
Western blot (WB) analysis of LC3 and p62 protein levels in 1.4E7 cells treated with GA. Cells were treated with 0–3 mM GA for 24 h. Lysates from 1.4E7 cells were loaded onto 4–15% polyacrylamide gradient gels (Ten μg of proteins per lane). This analysis was performed to evaluate cell viability in three independent experiments. (**a**) Proteins on PVDF membranes were targeted with anti-LC3, anti-p62, and anti-GAPDH antibodies. GS: gray scale. Gray scale change: the value of the gray scale was 5000 in Fusion FX software. WB analysis was performed in three independent experiments. GAPDH was used as the loading control. I and II show the LC3-I and -II position. (**b**) Protein levels of LC3-I were normalized to those of GAPDH. (**c**) Protein levels of LC3-II were normalized to those of GAPDH. (**d**) Protein levels of p62 were normalized to those of GAPDH. (**b**–**d**) Data are shown as the mean ± S.D. (*n* = 3). *p*-values were based on Tukey’s test. ** *p* < 0.01 vs. 0 mM GA. ^##^
*p* < 0.01 vs. 1.5 mM GA. ^$$^
*p* < 0.01 vs. 2 mM GA.

**Figure 3 nutrients-14-00332-f003:**
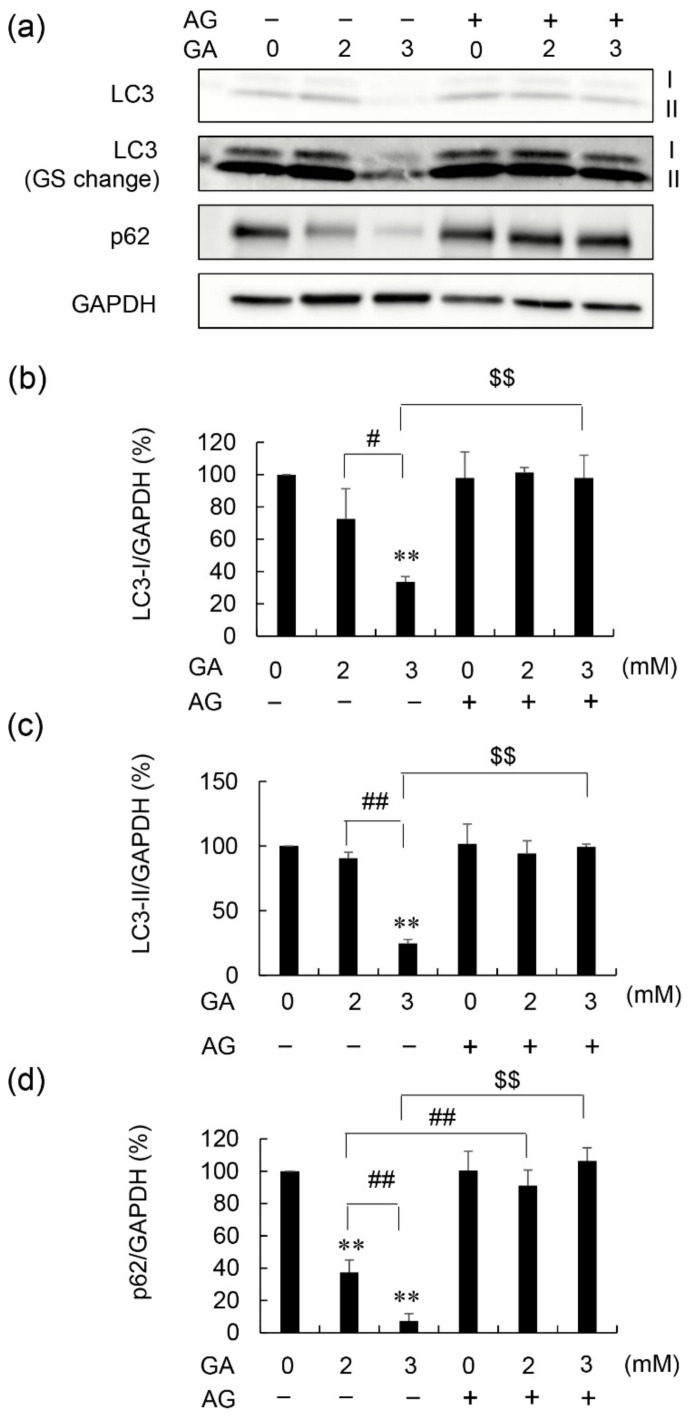
Detection of LC3 and p62 via WB analysis in 1.4E7 cells treated with GA and AG. 1.4E7 cell lysates were loaded onto from 4 to 15% polyacrylamide gradient gels (Ten μg of proteins per lane). Cells were initially treated with 0 or 16 mM AG for 2 h, followed by 0–3 mM GA for 24 h. This analysis was performed to evaluate cell viability in three independent experiments. (**a**) Proteins on PVDF membranes were targeted with anti-LC3, anti-p62, and anti-GAPDH antibodies. GS: gray scale. Gray scale change: the value of the gray scale was 10,000 in the Fusion FX software. I and II show the LC3-I and -II position. WB was performed in three independent experiments. GAPDH was used as a loading control. (**b**) Protein levels of LC3-I were normalized to those of GAPDH. (**c**) Protein levels of LC3-II were normalized to those of GAPDH. (**d**) Protein levels of p62 were normalized to those of GAPDH. (**b**–**d**) Data are shown as the mean ± S.D. (*n* = 3). *p*-values were based on Tukey’s test. ** *p* < 0.01 vs. 0 mM GA without AG. ^#^ *p* < 0.05 vs. 2 mM GA without AG. ^##^ *p* < 0.01 vs. 2 mM GA without AG. ^$$^ *p* < 0.01 vs. 3 mM GA without AG.

**Figure 4 nutrients-14-00332-f004:**
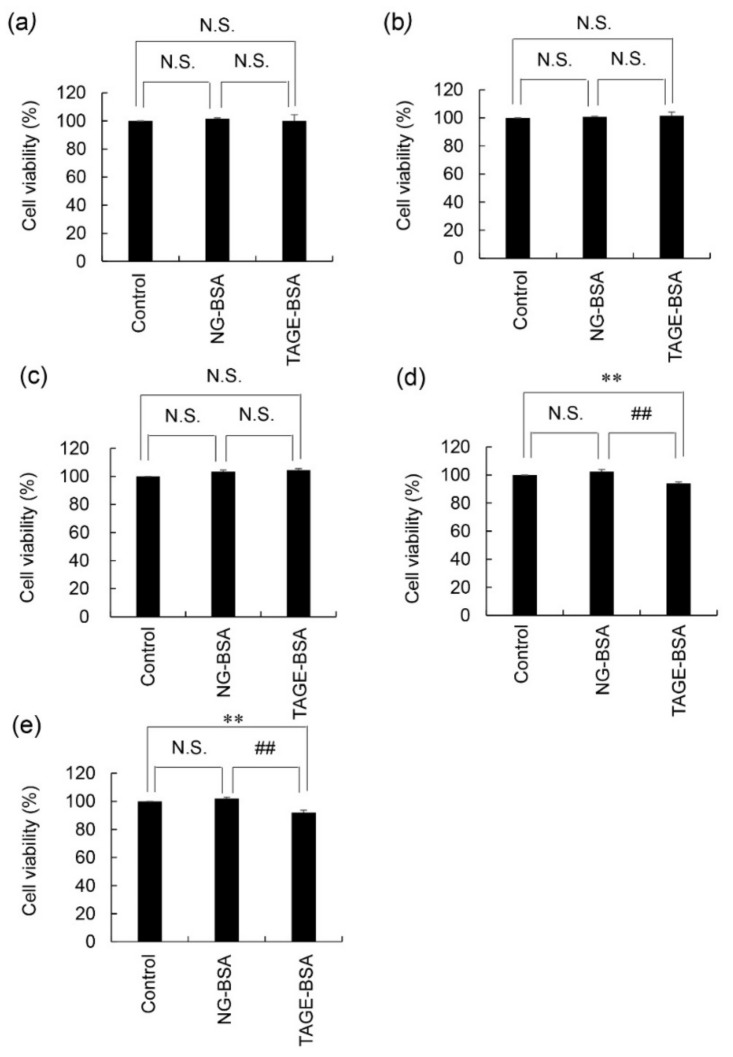
Cytotoxicity of TAGE-BSA against 1.4E7 cells with assessed using the CellTiter-Glo luminescent assay. Cells were seeded in five wells and the average was calculated. This assay was performed to evaluate cell viability in three independent experiments. Data are shown as the mean ± S.D. (*n* = 3). *p*-values were based on Tukey’s test. (**a**) Cells were treated with 10 μg/mL NG-BSA and TAGE-BSA for 24 h. (**b**) Cells were treated with 20 μg/mL NG-BSA and TAGE-BSA for 24 h. (**c**) Cells were treated with 50 μg/mL NG-BSA and TAGE-BSA for 24 h. (**d**) Cells were treated with 100 μg/mL NG-BSA and TAGE-BSA for 24 h. (**e**) Cells were treated with 200 μg/mL NG-BSA and TAGE-BSA for 24 h. (**d**,**e**) ** *p* < 0.01 vs. the control. ^##^
*p* < 0.01 vs. the NG-BSA treatment. (**a**–**e**) N.S.: Non significance.

## Data Availability

The data presented in this article are available upon request from the corresponding author.

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
