# Peer review of "Intracellular Toxic Advanced Glycation End-Products in 1.4E7 Cell Line Induce Death with Reduction of Microtubule-Associated Protein 1 Light Chain 3 and p62"

_nutrients, 2022, doi:10.3390/nu14020332_

Round 1
Reviewer 1 Report
In their manuscript Takata and coauthors set up to investigate the role of advanced glycation end-products (AGEs) in beta cell homeostasis. Although the topic is relevant, the manuscript presents many limitations.
Major points:
- The introduction session is very limited. The authors focus on their previous work on the effect of AGEs as biomarker in T2D and in citotoxicity, but they miss to cite several previous reports on AGEs role in pancreatic beta cell function and viability widely present in literature.
- Authors show that the production of intracellular AGEs induce pancreatic beta cell death, however an additional method of apoptosis detection is needed in order to confirm this observation, such as detection of cleaved caspase 3 for instance.
- The main conclusions are drawn conducing experiments in a cell line with tumourigenic backgrownd. These observations should be confirmed using primary islet cells.
- Since the effect of AGEs on pancreatic beta cells has been already described previously, the main novelty of this manuscript is represented by the studies on the role of autophagy in this phenomenon. However, further studies using modulators of autophagy must be included in order to corroborate the observations and define the pathway involved.
- The effect of extracellular AGEs is very mild and only observed under non-physiological concentrations. However it would be interesting to assess their effect in combination to environmetal stressors such as citokines and high fat exposure, and their effects on beta cell function.
Minor points:
- The titles and text of the Results session paragraphs must be reviewed, indeed they should provide an interpretation of the data shown, and not a mere read-out of the data already shown in the graphs.
Author Response
We would like to thank the reviewers for their comments on our manuscript. We
Point-by-point responses to the comments of Reviewer #1
In their manuscript Takata and coauthors set up to investigate the role of advanced glycation end-products (AGEs) in beta cell homeostasis. Although the topic is relevant, the manuscript presents many limitations.
Major points:
- The introduction session is very limited. The authors focus on their previous work on the effect of AGEs as biomarker in T2D and in citotoxicity, but they miss to cite several previous reports on AGEs role in pancreatic beta cell function and viability widely present in literature.
Answer 1. We discuss the generation of methylglyoxal-modified protein (MGO-AGEs) in pancreatic islets β-cells (β-cells) and the induction of cell death, increased levels of reactive oxygen stress, and inhibition of glyoxalase-I (New Ref. 6; PMID 28627583) in the Introduction and Discussion sections (Line 36-37 on the page 1, Line 300-304 on the page 8. The sentence is highlighted by Yellow background).
We also describe the possibility that MGO-AGEs in pancreatic islet β-cells inhibited caspase-3 activity following treatment with methylglyoxal (New Ref. 19; PMID 32384625). (Line 305-308 on the page 8. The sentences is highlighted by Yellow background).
Conversely, we discuss that dietary AGEs increased the levels of Nε-carboxymethyllysine (CML) in the blood, the accumulation of CML in the islets, insulitis, and the upregulation of RAGE protein in the islets (New Ref. 7; PMID 29157116) in the Introduction and Discussion sections. (Line 37-38 on the page 1, Line 308-310 on the page 8). The sentence is highlighted by Yellow background.
2. Authors show that the production of intracellular AGEs induce pancreatic beta cell death, however an additional method of apoptosis detection is needed in order to confirm this observation, such as detection of cleaved caspase 3 for instance.
Answer 2. We analyzed the protein levels of caspase-3 using a PVDF membrane, which was also used to analyze the levels of LC3, GAPDH, and p62. We used an anti-caspase-3 antibody (#9665S, Cell Signaling Technology), which can detect pro-caspase-3 and cleaved caspase-3. The results are provided in “Figure S8 for Reviewer only.” We observed bands of high molecular weight relating to caspase-3 (75 kDa). High molecular-weight caspase-3 may represent a TAGE-modified caspase-3, and TAGE-modification may affect the activity of caspase-3. Conversely, we were unable to detect cleaved caspase-3 (16–18 kDa). Although no positive control was used in this experiment, we thought that the detection of cleaved caspase-3 via western blot analysis may be difficult, as in PANC-1 (pancreatic ductal cell line) (Ref. 15). Attempts to confirm (1) TAGE-modification of caspase-3; (2) the effects of TAGE-modification on the activity of caspase-3; and (3) the relationships between cleaved caspase-3 and TAGE-modified caspase-3 would require time, but would enable us to obtain important data in a future study.
Based on your comment, we have mentioned the “importance of research on the type of cell death (e.g., apoptosis, necrosis)” in the Limitations section of the Discussion. (Line 395 on the page9 – Line400 on the page 10). The sentence is highlighted by Yellow background.
3. The main conclusions are drawn conducing experiments in a cell line with tumourigenic backgrownd. These observations should be confirmed using primary islet cells.
Answer 3. We have changed the title of this article, as follows: “Intracellular Advanced Glycation End-Products in 1.4E7 cells induced death with downregulation of microtubule-associated protein 1 light chain 3 and p62.”
We add/rewrote in the conclusion of Abstract and Discussion sections (Line 23 on the page1, and Line 407 on the page 10).
Moreover, we clarify the use of a pancreatic islet cell line rather than primary islet cells in the Limitations section of the Discussion. (Line 389-395 on the page 9). These sentences are highlighted by Yellow background.
4. Since the effect of AGEs on pancreatic beta cells has been already described previously, the main novelty of this manuscript is represented by the studies on the role of autophagy in this phenomenon. However, further studies using modulators of autophagy must be included in order to corroborate the observations and define the pathway involved.
Answer 4. We previously described the possibility that autophagy is inhibited in rat primary cardiomyocytes, which generated intracellular TAGE (Ref. 11). Although data are only provided for LC3 and p62 in Ref. 11, we analyzed HSP90 and HSP70, which are associated with autophagy and protein degradation. We observed that high molecular weights of HSP90 and HSP70 were generated in rat primary cardiomyocytes, which generated intracellular TAGE (Figures S9 and S10 for Reviewer Only). Recently, we attempted to identify TAGE-modified proteins associated with autophagy. Although further studies using modulators of autophagy are needed, there remains a possibility that target proteins are TAGE-modified, which represents a significant problem.
Therefore, we have discussed the need for further studies in the Limitations section of the Discussion. (Line 400-405 on the page 10). The sentence is highlighted by Yellow background).
5. The effect of extracellular AGEs is very mild and only observed under non-physiological concentrations. However it would be interesting to assess their effect in combination to environmetal stressors such as citokines and high fat exposure, and their effects on beta cell function.
Answer 5. To address your comment, we have included descriptions of such studies in the section of Discussion (Line 378-388 on the page 9). The sentence is highlighted by Yellow background.
In LGALS3-/- mice, with knockdown of galectin-3, which is the receptor for degrading AGEs, serum levels of IL-1β increased, the area of IL-1β in the pancreatic islets increased, and insulitis was promoted (New Ref. 13; PMID 23349493). In C57/BL6 mice, the intake of both the dietary AGEs and a high-fat diet induced an increase in the serum levels of CML, degradation of isles, and insulin-producing cell displacement from the periphery to the center of the islets (New Ref. 37; PMID 16046296).
Minor points:
- The titles and text of the Results session paragraphs must be reviewed, indeed they should provide an interpretation of the data shown, and not a mere read-out of the data already shown in the graphs.
Answer 1. We have provided an interpretation of our data in the section of Results. Please read the sentences in the section of Results (page 3-6) which are highlighted by Yellow background).

Reviewer 2 Report
There are minor spelling errors.
Discussion needs should be more connected with the results.
In the graphs it is not clear which groups are being compared. It would be helpful if all graphs are presented like the graph a and b of figure 4.
the full western blots do not have ruler marker. We can not be sure of the molecular weights reported unless a protein ladder is run with the samples. This is a red flag.
There are more than 20 self citations.
Author Response
We would like to thank the reviewers for their comments on our manuscript. We have addressed the comments provided and hope that our responses are satisfactory. Our point-by-point responses to each reviewer are listed below.
Point-by-point responses to the comments of Reviewer #2
Comments and Suggestions for Authors
1. There are minor spelling errors.
Answer 1. We have corrected the spelling errors, and the manuscript has been reviewed for English language by Editage. More, we checked our article with iThenticate software, the duplication rate of sentences were researched. The duplication rate with single article is less than 7% and the total duplication rate is 29%. (without References).
2. Discussion needs should be more connected with the results.
Answer 2. The discussion has been revised to ensure it reflects the results presented in the manuscript. Please read Line 325-330 on the page 8, Line 352 and 357-358 on the page 9. These sentences are highlighted by Yellow background.
3. In the graphs it is not clear which groups are being compared. It would be helpful if all graphs are presented like the graph a and b of figure 4.
Answer 3. We have included a “bar and symbol” to indicate significant differences between two groups in the Figures 1-3. These are used to demonstrate the relationships between two groups (e.g., second group versus the fifth group). However, a symbol has been used to indicate a significant difference compared with the “control (the first group)” as well as in the previous graph.
4. The full western blots do not have ruler marker. We can not be sure of the molecular weights reported unless a protein ladder is run with the samples. This is a red flag.
Answer 4. In the images showing the full blots for GAPDH and p62, no HRP-linked molecular weight markers were observed because these were removed by the western blot stripping solution. We have provided full images of the PVDF membranes. LC3, GAPDH, and p62 were analyzed using the same PVDF membrane. In addition, we have provided images of the PVDF membrane under visible light (under the White Epi mode). The lanes containing molecular weight markers that can be observed under visible light and HRP-linked molecular weight size markers are included. The positions of LC3-I and LC3-II were determined using an HRP-linked molecular weight marker. The positions of GAPDH and p62 were determined using a molecular weight marker, which can be observed under visible light. The bands were obtained using a primary antibody, a secondary antibody, and reagents of the ImmunoStar LD Kit. The bands of these compounds could be observed under visible light. The position of bands for GAPDH or p62 are shown, because the bands obtained under the Chemilumi mode and White Epi mode were matched.
Please see Figure S1-7. However, Figure S8-10 are reviewer only.
5. There are more than 20 self citations.
Answer 5. We have removed 9 such references in the updated version of the manuscript.

Round 2
Reviewer 1 Report
Although in their revised manuscript Takata and colleagues addressed some of the points of concern by mainly editing the text, the lack of additional experiments and results makes the manuscript scarcely novel or interesting. Therefor, my advice is to improve the study with further experiments to clarify the role of TAGEs on beta cell viability and function.
Reviewer 2 Report
The authors have made suggested changes